# Analysis of the Different Scenarios of Coach’s Anger on the Performance of Youth Basketball Teams

**DOI:** 10.3390/ijerph19010459

**Published:** 2022-01-01

**Authors:** Victor Hugo Duque, Pedro Saenz-López, Miguel Ángel Gómez-Ruano, Sergio J. Ibáñez-Godoy, Cristina Conde, Bartolomé J. Almagro, José Antonio Rebollo

**Affiliations:** 1Grupo de Investigación Prácticas Corporales, Sociedad, Educación-Currículo (PES), Instituto Universitario de Educación Física, University of Antioquia, Medellin 050010, Colombia; victor.duque@udea.edu.co; 2Faculty of Education, Psychology and Sport Sciences, COIDESO, Universidad de Huelva, 21071 Huelva, Spain; psaenz@uhu.es (P.S.-L.); almagro@dempc.uhu.es (B.J.A.); joseantonio.rebollo@dempc.uhu.es (J.A.R.); 3Faculty of Sport Science, Polytechnic University of Madrid, 28040 Madrid, Spain; miguelangel.gomez.ruano@upm.es; 4Training Optimization and Sports Performance Research Group (GOERD), Sport Science Faculty, University of Extremadura, 10003 Cáceres, Spain; sibanez@unex.es

**Keywords:** negative emotions, basketball, anger, competition, youth

## Abstract

In spite of the negative effects of anger, coaches are often seen becoming angry during games. This is especially worrying in U18 categories. Thus, the objective of this study was to identify the influence that the coach’s anger has on the performance of a basketball team in competition. For this, an ad hoc observation tool was designed, in which 587 moments of anger from the coaching staff (64 coaches) were recorded in the 24 semi-final and final matches of the Spanish Autonomous Region Team Championships in 2019 and 2020 in the *infantil* (M = 14 years old) and *cadete* (M = 16 years old) categories. The results show that, in response to most incidents of coach anger, the performance of the team did not change. Significant differences were identified in some scenarios, with low- or medium-intensity anger targeted at the defence, where the team performance improved. However, anger towards the referee in the last quarter with scores level had a negative influence on the team’s performance.

## 1. Introduction

Basketball is a collective, competitive sport with considerable social impact in Spain, which is why it triggers various emotions among its participants [1]. Depending on the significance and intensity, these emotions are expressed quite frequently through aggressive behaviour [2]. This energy caused by emotional impulses sometimes induces a certain activation and desire to improve, and in many other cases blocks, inhibits, paralyses, and leads to feelings of failure, low self-esteem, and negative attitudes [3]. Coaches play a fundamental role in this regard, due to the emotional contagion that exists between the leader’s mood and the climate generated among the members of a team [4]. Therefore, in addition to planning training sessions and leading the team in competitions, coaches need to develop and maintain motivation in athletes, fostering an emotional climate that engages the team and facilitates learning and goals [5,6,7,8]. In the educational field, it has been shown that the teacher’s attitude creates an emotional climate that can facilitate or hinder the achievement of the proposed objectives [9].

In competition, the positive emotions of one team’s members can influence the failure of those in the other [10]; likewise, the emotions shown by the coach will affect the emotional state of the players—positively or negatively—and the resulting behaviour influences team cohesion and performance [11]. Many coaches are unaware of the negative effect that their attitude can have on athletes, with consequences that affect their motivation, decision making, anxiety, confidence, and even aspects of their physiology and, therefore, their performance [12]. Perhaps the most frequently observed negative emotion in coaches is anger, which according to Izard [13] is a primary reaction resulting from preventing an individual from achieving a targeted goal or satisfying a need. Spielberger et al. [14] define anger as an emotional state that varies in intensity from irritation to rage, and is expressed internally and/or externally. The former includes angry feelings and thoughts without being overtly expressed, while the second is communicative, sometimes with non-threatening expressions, and at other times aggressive and intended to hurt. For [15], the level of anger is notable, and varies according to the level of the league in which the athletes participate. This emotion, which according to Smith [16] has a characteristic facial expression, has been studied in sports, with various objectives—for example, the relationship of anger with aggressive behaviour [17], with the athlete’s internal dialogue [18], with their physiological responses [19], with sports performance [20,21], and with the level of competition [22]. In football, it was shown that the teams that played with more anger scored fewer goals, while the teams that played with more happiness and confidence scored more goals, in a world championship [23]; this was confirmed by the authors of [24], who showed that the teams that showed more aggressiveness achieved a worse performance.

In basketball, anger has predicted how much negative performance influences team behaviour at the collective level [25], manifesting itself more often in lost matches [26]; however, in sports such as rugby, it has been associated with higher performance levels [27], perhaps because they require a greater amount of energy [28], as occurs in contact or combat sports disciplines, such as ice hockey, karate, and boxing [21,29]. Thus far, sporting performance in basketball has been measured based on the statistics of the matches, such as percentage of shots, rebounds, or assists [30], or physical variables such as internal or external load [31]. It would be wise and original to find data on the influence of emotions, such as the influence of the coach’s emotional state on the score in basketball competitions. Based on the above, our hypothesis is that the coach’s anger during the game negatively influences the team’s score; therefore, the main objective of this study was to identify the influence that the coach’s anger has on the performance of the team in competition during each match played.

## 2. Methods

### 2.1. Design of the Study

This study is framed in descriptive observational research, in which the behaviours to be analysed were recorded, as they developed, in a natural context without the intervention of the researcher [32].

### 2.2. Participants

There were n = 587 moments of anger from 64 coaches and technical assistants in 24 semi-final and final matches of the Spanish Autonomous Region Team Basketball Championships for School Age (CESA), held in the province of Huelva (Andalusia, Spain) in the years 2019 and 2020 in the infantil (13 and 14 years of age) and cadete (15 and 16 years of age) categories, for both male and female teams.

### 2.3. Measures

An ad hoc observation sheet was designed to identify the moments in which the coaches and technical assistants expressed anger during the game, in which the following variables were observed. A similar tool has been used in previous research [33]:(a)Period;(b)Minute;(c)Score;(d)Situation—1: attack; 2: defence; 3: time-out; 4: rest; 5: others;(e)Coach per team: E1 and E2; assistant per team: A1 and A2;(f)Intensity—1: low (an angry gesture or tone of voice); 2: medium (when the level of anger causes the team to perceive it clearly); 3: high (when the gesture or tone shows aggressiveness);(g)Cause—1: attack situation; 2: defence situation; 3: referee’s decision; 4: others;(h)At whom—1: team on the field; 2: team on bench; 3: specific player; 4: referees; 5: control table; 6: others;(i)Observations: to describe the play or identify relevant aspects that complement the information.

The sheet identified the match data (year, category, gender, teams, etc.).

Subsequently, the matches were observed on video, identifying each anger event recorded live to complete the observation sheet with the following data:(j)Accumulated anger: 1 if cumulative; 0 if not cumulative;(k)Points scored and conceded in the 3 previous and subsequent possessions in attack and defence, taking the score carried over. For example, after the anger event we wrote down the points score in the first attack possession and the points conceded in the first defence possession [33].

### 2.4. Procedure

Initially, a pilot was carried out in a tournament for the U14 and U16 categories, in which the researchers observed two live basketball games, recording the data in the observation tool, which served to precisely define and screen the criteria. Subsequently, during the group phase of the Spanish Championship, observations were made for training, choosing matches including the teams that had qualified for the semi-finals, so that some traits of the coaches and technical assistants on the game management were also identified. In the first quarter of the game, the researchers collected data together and agreed on the reasons why the moment of anger was recorded. During the second quarter, each researcher took records individually, and at the end of the quarter, they met to compare one another’s data, identifying and discussing the discrepancies in the recorded events. In the 3rd and 4th quarters of the game, the procedures for the 1st and 2nd quarters were applied again. This process was repeated for two games, going from 70% intercoder agreement in the first data collection to more than 90% for the last three records, so the agreement among observers was high. This training was carried out by two different researchers in each game.

In the semi-finals, each researcher recorded data from different matches live on the instrument, and in the four final matches, two matches were taken individually and two matches jointly by two researchers. Subsequently, the video of each game was observed in detail, thus completing the variables of points scored and conceded and accumulated anger in the possessions before and after the coaches’ moments of anger. The official videos were downloaded using the Mozilla Firefox browser’s Video DownloadHelper Plugin from the contest’s official media page, with prior permission from the Basketball Federation.

### 2.5. Data Analysis

First, a two-stage cluster analysis was carried out to try to differentiate the types of episodes of anger based on the factors or contexts in which they occurred (e.g., period of the game, score difference, intensity, cause, accumulation of anger, whether there were changes due to the anger, whether a time-out was called, and which coach was the one who got angry). The model enables exploration of the best classifications based on categorical and continuous variables using Schwartz’s Bayesian information criterion (BIC) and the silhouette measure. Second, the normality (Shapiro–Wilk) tests were performed, finding that all the variables responded to non-normal distributions (*p* < 0.05). Third, repeated-measures comparisons (Wilcoxon test) were carried out for the variables points scored and conceded before and after the anger episode (1, 2, and 3 ball possessions) in each cluster. Fourth, the differences in points scored and conceded (before vs. after the anger episode) between the 6 clusters were compared using the Kruskal–Wallis test. The Bonferroni test was used for pairwise comparisons. All analyses were performed with the IBM SPSS statistical package for Macintosh version 25.0 (IBM. Corp., Armonk, NY, USA). The level of significance was established at *p* < 0.05.

## 3. Results

The results of the two-stage cluster analysis (see Table 1) showed the classification of the coaches’ anger episodes into six groups (scenarios or contexts or situations). The model was significant, with good values for the silhouette value (>0.5), highlighting the importance of five factors (i.e., the period of the game, the intensity of the anger, the cause, the accumulation of anger, and the difference in the score) to clarify the episodes (variables with importance values greater than 0.1 within the model).

The six episodes found can be characterised as follows (in order of % occurrence): C1 is the context that occurs most, prevailing in the second quarter of the game, during defensive actions, with accumulated anger of low intensity when the scores are level (−4 to +6 points); C6 is the second most common context, which occurs in the first quarter of the match, as a single episode of low intensity, mainly due to the referee and when the scores are level; C4 is the context that occurs in the fourth quarter of the game as a single episode of medium intensity, linked to defence, and ends with scores level; C5 represents episodes that occur mainly in the third quarter of the game as single low-intensity episode of anger due to the referee, when the game is being lost by between 7 and 20 points; C2 represents episodes that occur in the second quarter of the match as a single episode of low-intensity anger towards the referee, when the scores are level; and C3 represents low-intensity anger episodes, in the third quarter of the game, constituting only anger towards the referee, and when the team is losing or winning by more than 20 points, or when the scores are level.

Table 2 presents the descriptive results of points scored and conceded one, two, and three ball possessions before and after the episode of anger.

The results of repeated-measures comparisons (see Table 2; Wilcoxon test) for points scored and conceded before and after the episode of anger (one, two, and three ball possessions) showed the following statistically significant differences: (1) cluster 1 scoring more points in one ball possession (*p* = 0.02) and three ball possessions (*p* = 0.02) after the episode of anger; (2) cluster 3 receiving fewer points after anger in one ball possession (*p* = 0.01); (3) cluster 4 receiving fewer points in one ball possession after the episode of anger (*p* = 0.02); and (4) cluster 6 receiving more points against in three ball possessions (*p* = 0.045) and scoring more points in one ball possession (*p* = 0.02).

## 4. Discussion

The objective of this study was to identify the influence that the basketball coach’s anger has on the performance of a team in competition. For this, the points scored and conceded during the game in the three possessions in attack and defence before and after the anger were analysed. To analyse the influence of other variables—such as the situation, the cause, the intensity, or at whom the anger was directed—the events were classified into six scenarios that show different trends in sports performance.

In the most frequent scenario, cluster 1, the results show that after the accumulated low-intensity anger of the coaches when the scores are level (−4 to +6 points) while the team is defending, the team scores more points in the first and third possessions. Finding the right activation level is essential to achieving effective performance [34]. Anger is a source of energy [35], and its effectiveness will depend on the control of actions in relation to the proposed objective [3]. This may explain why low-intensity anger in defensive actions can activate some players. Something similar occurs in cluster 4, in which single bouts of medium-intensity anger in the fourth quarter—also related to defence—are somewhat effective, in that the team concedes fewer points in the first post-anger defensive possession. This increase in defensive activation in the players in a controlled manner can have a positive impact on performance—something that the coach should take into account when directing the games.

The second most frequent scenario was cluster 6, which occurs in the first quarter, with scores level, and the anger is of low intensity and directed at the referee. In these situations, teams score more points in the first possession and, conversely, concede more points in the subsequent three possessions. Anger is an emotion that provides energy and, depending on the cause, can generate anxiety and frustration [36,37], which can have a negative effect on the concentration and confidence of athletes—particularly in the face of external causes [38], such as a refereeing decision. Zur, Cooke, Woodman, Neil, and Udewitz [39], in their study of elite fencing athletes, found very similar results; initially, anger improved the reaction time of athletes, but subsequently reduced their performance. The results show that, contrary to what the coaches may believe, the referees are not influenced by the comments of the coaches; furthermore, they can have a negative effect on performance. According to the above, it is preferable to avoid protesting to the referee, no matter how minimal it may be.

Protests directed at referees with a certain level of anger are a constant in sport at any level [40]; however, as in the situation described above, previous studies show that unsportsmanlike techniques and fouls given against a team cause worse subsequent performance [41]. Along these lines, Ring, Kavussanu, Al-Yaaribi, Tenenbaum, and Stanger [41] showed that anger damages performance, among other causes, due to the distraction it creates with respect to the sporting objective. Emotions are contagious [42]; thus, in situations of tension between the coach and the referees, the negative effects of anger could influence the team’s performance. In fact, anger leads to a negative internal dialogue in the athlete that damages their performance [18]. Uphill et al. [26] showed that anger and shame are the two emotions that are most related to failure in basketball.

In the situation of cluster 5, which occurs mainly in the third quarter with single low-intensity episodes of anger directed at the referee, and when the team are losing by between 7 and 20 points, the differences were not significant. The same occurred in the context of cluster 2, which occurs in the second quarter with a single episode of mild anger directed at the referee when the scores are level. This once again shows that angry protests to the referees are ineffective, as the performance remains the same or decreases. Different groups of athletes take part during a match. The referees participate actively in the game, showing that the pressure exerted on them does not cause them to change their criteria, since there are other factors that affect their stress [43].

The differences in points scored and received in the three offensive and defensive possessions before and after anger were not significant. As described by Bisquerra [35], anger is a basic emotion that appears, essentially, when things do not happen as we would like, or when we consider someone to be treating us badly. In sport, this usually occurs when an individual makes an assessment of the situation and it is far from their objectives or expectations [44]. Therefore, in basketball, the coach’s anger is usually caused by frustration that the game is not going as intended, or by some refereeing decision, and only occasionally when someone treats them badly. In fact, the appearance of anger in sport is more frequent when perceived self-esteem is low [45]. These data indicate the need to be aware of anger, reflecting on its cause(s), and especially on the behavioural reaction. It was observed in this study that, in general, anger is ineffective. Therefore, it is recommended that coaches train in emotional awareness and control, so as to prevent episodes of anger from negatively affecting the performance of their team.

The natural reaction to anger is usually one of irritation and fury shown through gestures, tones, insults, or aggressions [35]. In this sense, the number of high-intensity anger events did not produce significant values. González-García et al. [46], concluded that the greater the manifestation of anger, the lower the levels of competence in the athletes. At the collective level, García-García et al. [24], showed that aggressiveness negatively influenced performance in football.

This study has some limitations, such as the sample focusing on a single championship, or the difficulty of externally observing an emotion such as anger. However, the importance of emotional education in coaches makes further research on this topic worthwhile. Therefore, it would be necessary to analyse the effects of anger in adults and professional teams, with audio included, in order to analyse the coaches’ language, as well as to understand the players’ perceptions of their coaches’ anger.

## 5. Conclusions

In conclusion, this study shows that, in most situations of coach anger, the performance of the team does not change. Significant differences only appear in a scenario with low- or medium-intensity anger aimed at the defence, in which the team’s performance improves. However, anger towards the referee in the last quarter with scores level has a negative influence on the team’s performance.

Based on these results, coaches should be encouraged to train in emotional skills to manage anger with awareness and effectiveness. In training, it should be recommended that coaches take part in the education of the players, including emotional aspects such as anger management. In addition, generating a positive emotional climate in athletes through the example of the coach will help to achieve this goal of improving commitment, and probably to achieve a better performance [23,47,48].

In high-level competition, low- and medium-intensity anger could be managed, provided they produce positive results. Even so, it has been observed that, in recent decades, elite coaches have shown ever greater control of negative emotions. In any case, it is necessary to carry out studies in different contexts that delve deeper into this issue.

## Figures and Tables

**Table 1 ijerph-19-00459-t001:** Cluster obtained via the two-stage method for the contextual variables related to the episode of anger (values in % frequency).

		C1	C2	C3	C4	C5	C6
		N = 120 (22.1%)	N = 80 (14.7%)	N = 77 (14.2%)	N = 85 (15.6%)	N = 82 (15.1%)	N = 100 (18.4%)
Variables	BIC	6233.2	5967.9	5708.4	5479.4	5305.3	5146.9
Period (I = 1.0)	1°	15.0	0	0	35.3	31.7	51.0
	2°	45.8	100	3.9	12.9	0	15.0
	3°	31.7	0	64.9	7.1	39.0	22.0
	4°	7.5	0	31.2	44.7	29.3	12.0
Intensity (I = 0.54)	Low	55.8	65.0	46.8	0	73.2	100
	Medium	41.7	22.5	53.2	92.9	20.7	0
	High	2.5	12.5	0	7.1	6.1	0
Cause (I = 0.35)	Attack	18.3	18.8	14.3	28.2	12.2	24.0
	Defence	44.2	20.0	58.4	43.5	17.1	36.0
	Referee	32.5	56.3	22.1	22.4	64.6	37.0
	Others	3.3	5.0	3.9	3.5	6.1	2.0
	Two episodes	1.7	0	1.3	2.4	0	1.0
Episodes of anger (I = 0.94)	Single	0	96.3	100	69.4	61.0	100
	Acumulated	100	3.7	0	30.6	39.0	0
Score difference (I = 0.811)	Losing > 21	2.5	1.3	26.0	0	0	0
	Losing 7–20	19.2	13.8	11.7	0	53.7	18.0
	Winning 5–19	10.8	26.3	10.4	16.5	46.3	3.0
	Winning > 20	15.0	8.8	26.0	0	0	0
	Level −6 to 4	52.5	50.0	26.0	83.5	0	79.0

**Table 2 ijerph-19-00459-t002:** Descriptive results of points scored and conceded for each cluster, and mean differences between performance before and after the episode of anger (Wilcoxon repeated-measures test).

		Before		After			
	N	M	DT	M	DT	Z	*p*
Cluster 1							
Scoredpoints1PB	120	0.52	0.91	0.66	0.97	−1.19	0.23
Scoredpoints2PB	120	0.97	1.17	1.37	1.44	−2.25	0.02 *
Scoredpoints3PB	120	1.44	1.67	1.95	1.74	−2.39	0.02 *
Concededpoints1PB	120	0.98	1.08	0.95	1.08	−0.29	0.77
Concededpoints2PB	120	1.77	1.62	1.58	1.31	−0.94	0.35
Concededpoints3PB	120	2.50	2.00	2.38	1.74	−0.39	0.69
Cluster 2							
Scoredpoints1PB	80	0.64	1.12	0.79	1.10	−0.80	0.42
Scoredpoints2PB	80	1.40	1.72	1.60	1.51	−0.67	0.50
Scoredpoints3PB	80	2.13	2.10	2.45	1.97	−0.88	0.38
Concededpoints1PB	80	0.84	1.19	0.81	1.08	−0.25	0.80
Concededpoints2PB	80	1.56	1.45	1.55	1.62	−0.01	0.99
Concededpoints3PB	80	2.14	1.77	2.21	1.90	−0.37	0.71
Cluster 3							
Scoredpoints1PB	77	0.51	0.95	0.44	0.82	−0.52	0.60
Scoredpoints2PB	77	1.09	1.33	0.94	1.24	−0.88	0.38
Scoredpoints3PB	77	2.03	1.78	1.68	1.63	−1.28	0.20
Concededpoints1PB	77	1.01	1.13	0.64	0.87	−2.48	0.01*
Concededpoints2PB	77	1.52	1.40	1.40	1.31	−0.80	0.62
Concededpoints3PB	77	2.14	1.79	2.08	1.64	−0.40	0.93
Cluster 4							
Scoredpoints1PB	85	0.61	1.05	0.71	1.11	−0.50	0.62
Scoredpoints2PB	85	1.19	1.43	1.42	1.51	−0.95	0.34
Scoredpoints3PB	85	1.93	1.86	1.94	1.76	−0.04	0.97
Concededpoints1PB	85	1.13	1.17	0.76	1.02	−0.34	0.02 *
Concededpoints2PB	85	1.73	1.52	1.56	1.46	−0.73	0.43
Concededpoints3PB	85	2.35	1.85	2.44	1.64	−1.93	0.69
Cluster 5							
Scoredpoints1PB	82	0.72	1.07	0.63	1.02	−0.43	0.67
Scoredpoints2PB	82	1.27	1.45	1.26	1.41	−0.07	0.95
Scoredpoints3PB	82	1.99	1.82	1.90	1.68	−0.49	0.62
Concededpoints1PB	82	0.99	1.27	1.11	1.22	−0.34	0.74
Concededpoints2PB	82	1.79	1.84	1.98	1.74	−0.73	0.46
Concededpoints3PB	82	2.30	2.09	2.73	2.18	−1.27	0.20
Cluster 6							
Scoredpoints1PB	100	0.49	0.95	0.87	1.17	−2.31	0.02 *
Scoredpoints2PB	100	1.15	1.37	1.53	1.55	−1.82	0.07
Scoredpoints3PB	100	1.76	1.56	2.14	1.81	−1.51	0.13
Concededpoints1PB	100	0.78	1.03	1.01	1.14	−1.47	0.14
Concededpoints2PB	100	1.40	1.41	1.66	1.63	−0.96	0.34
Concededpoints3PB	100	2.07	1.76	2.50	1.68	−1.93	0.05 *

Note: Concededpoints#PB and Scoredpoints#PB represent the points in the first, second, and third ball possessions. * *p* < 0.

## Data Availability

Data can be supply by the authors.

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
