# Peer review of "Analysis of the Different Scenarios of Coach’s Anger on the Performance of Youth Basketball Teams"

_ijerph, 2022, doi:10.3390/ijerph19010459_

Round 1

Reviewer 1 Report

An interesting observational study that will be of interested to researchers interested in the determinants of sports performance. Below are some comments for how the piece can be improved to make it suitable for publication:

Some sentences are quite long which makes them harder to follow. E.g. lines 60-71 is currently all one sentence.

The introduction could be structured more logically to make the gap that you are addressing with your research clearer. For example, you could start by talking about what anger is, followed its role in sports more generally, then basketball specific studies…

“Sports performance in basketball has been measured based on the analysis of the game (Ibáñez, García-Rubio, Rodríguez-Serrano, and Feu, 2019), or physical variables (Reina, García-Rubio and Ibáñez, 2020)…” Analysis of the game and physical variables are too vague, what physical variables were used?

How was the ad hoc observation tool designed? Is it based on a measure that has been used in previous research?

“In the semi-finals, each researcher recorded data from different matches live on the instrument, and in the four final matches, two matches were taken individually and two matches jointly by the researchers.” Why was this format used? It is not clear to me whether all matches were observed by two or more researchers (meaning that inter-rater reliability could have been conducted), or not.

‘Received’ is misspelt in Table 2. Related to this, what is the difference between points scored and points received? It is not clear. Make sure that you clearly define this as it seems to be central to your results.

The discussion could be improved by thinking about the limitations of the current work and future directions.

Reviewer 2 Report

The paper is in line with the objectives of the special issue. However, in my opinion, the paper presents several limitations that should be solved in order to be published.

The first problem concerns the originality of the data. The authors do not provide a sufficient argumentation of how the paper adds to the many studies in this area.

The introduction presents gaps in important research in the field of social and emotional regulation and its effects, particularly in the field of psychology.

The method has limitations inherent to its observational nature that should be taken into account and properly addressed. The authors are silent about the timings, the number of observers and the concordance matrix. It is very important that the methodology and procedure be completely clear, which is not the case at the moment.

Thus, all these issues I believe the first one is the most important one, because, for me is completely unclear how this study goes behind what is already done in other studies and how a limited study (in terms of methods) could be potentially interesting and important to the research field.

Round 2

Reviewer 2 Report

The author provided substantial revisions to the manuscript and addressed overall all my previous concerns about the initial version. I believe the authors improved substantially the paper and it is now, in my opinion, ready for publication.